# Optimization and Security in Information Retrieval, Extraction, Processing, and Presentation on a Cloud Platform

**Adrian Alexandrescu** 

Department of Computer Science and Engineering, Faculty of Automatic Control and Computer Engineering, "Gheorghe Asachi" Technical University of Iași, Iași 700050, Romania; aalexandrescu@tuiasi.ro

**Abstract:** This paper presents the processing steps needed in order to have a fully functional vertical search engine. Four actions are identified (i.e., retrieval, extraction, presentation, and delivery) and are required to crawl websites, get the product information from the retrieved webpages, process that data, and offer the end-user the possibility of looking for various products. The whole application flow is focused on low resource usage, and especially on the delivery action, which consists of a web application that uses cloud resources and is optimized for cost efficiency. Novel methods for representing the crawl and extraction template, for product index optimizations, and for deploying and storing data in the cloud database are identified and explained. In addition, key aspects are discussed regarding ethics and security in the proposed solution. A practical use-case scenario is also presented, where products are extracted from seven online board and card game retailers. Finally, the potential of the proposed solution is discussed in terms of researching new methods for improving various aspects of the proposed solution in order to increase cost efficiency and scalability.

**Keywords:** optimization; security; cost efficiency; information retrieval; information extraction; crawler; cloud services; vertical search engine; e-shopping; distributed systems

---

## 1. Introduction

The Internet contains a huge amount of ever-growing data that end-users must sift through in order to obtain the information that is relevant to them. When a user wants to find a specific product for online purchase, that person usually uses a general search engine or a specialized shopping search engine like Google Shopping, PriceGrabber, or Shopzilla [1]. Two of the first problems that occur when developing a shopping search engine is where you find the product information and how you retrieve that data. If those problems are solved, the next step is to use information retrieval techniques, extract the data, then perform a reverse indexing to obtain, for each word in the product names, a list of products that contain that word. Finally, the end result is to develop a search engine user interface so that one can search for a product by name. More complex shopping search engines allow searching depending on the features of the product (e.g., manufacturer name, dimensions, color), not just the product name.

Existing relevant research in this domain refer to vertical search engines [2,3], which are focused on a specific topic and generally use a targeted crawler, as opposed to crawling the "entire" web. One of the most common verticals is shopping, but we can go even further and consider the vertical to be shopping for a specific type of product (e.g., smartphones, power tools, clothes). When developing a shopping search engine, it helps to start with a specific type of product and then add more verticals to see if the provided solution is as efficient.

In terms of information retrieval, there are web crawlers and web scrapers [4]. A web crawler is an application that accesses webpages, extracts the URLs, and then repeats this process for each page

denoted by the extracted URLs. The process can be viewed as a breadth first traversal in a directed graph where each vertex represents a webpage and each edge represents a link from one page to another. On the other hand, a web scraper extracts data from webpages by identifying the useful information and then storing it in a database. The method of identifying the required information can be as trivial as specifying the HTML path to the data beforehand, or it can employ data mining techniques to ensure a fully automated retrieval process [5]. Crawling implies some scraping in the sense that the links have to be extracted, and scraping implies some crawling because, usually, the data is gathered from multiple webpages and the scraper needs to access those pages' content.

Finding the URLs that contain product data and the location of the product features on a webpage is very difficult due to fact that websites are very different in presenting their information. Usually, the research regarding this topic is proprietary and existing shopping search engines keep secret their techniques of extracting data. There is some public research in terms of structured data extraction [6,7], but the solutions are very general and their efficiency is questionable when it comes to product information extraction.

Due to the fact that the research presented in this paper follows a complex flow of information that passes through clearly distinct components, more related work is referred to in the sections presenting the proposed solution.

The main aim of the research is to propose a solution for having a vertical shopping search engine, from retrieving the product information from online retailers to offering an interface where the user can search for the desired products and can also view the price history of each product. This research is a continuation of the work from [8], where a framework for information retrieval, processing, and the presentation of data is presented. The main differences are streamlining the information flow between components, a better modularization, and especially, the multiple optimizations that were made to lower resource consumption and therefore lower the running costs. On top of this, a security layer was added to make the crawler run smoothly and to prevent third parties from exploiting the proposed system.

In the current paper, the proposed crawling and extraction components are a combination between traditional web crawling and web scraping, i.e., the proposed crawler is bound to specific websites, but it also finds new pages on those websites and retrieves them for extraction. Moreover, at this point, the crawling and extraction template is manually set for each website, and it is used to obtain the product features. In terms of the communication between the main components, this is achieved by means of web services, which are a very good method for sending data in heterogeneous environments because of the little volume of overhead information that is being sent, while having a flexible information transfer protocol structure [9].

Throughout the paper, the challenges that the considered environment entails, the chosen solutions, and possible improvements are identified. The main optimizations are: a novel method of representing the crawl and extraction configuration/template, several steps to lower cloud resource usage by keeping a separate product index, grouping multiple products in the same database entity, using a memory cache layer, and other methods for keeping the database reads and writes to a minimum while taking into account the limitations imposed by the cloud platform.

## 2. Proposed Solution

### 2.1. General Overview

The goal is to gather specific information and to make it easily accessible to the user in a cost-effective and secure manner. Therefore, this paper proposes a solution for information retrieval, extraction, processing, and the presentation of data that is optimized for efficiency and cost reduction. Throughout this paper, the specific information pertains to product details gathered from online retailers, and the whole process flow is essentially everything needed for a fully functional shopping search engine. On the other hand, the proposed solution is designed to be generic, which means

that it can be used to obtain different types of structured data from webpages, and not just product information. For example, it can be used for finding medication based on symptoms, recipes based on ingredients, or news articles referring to specific topics. A significant improvement compared to a traditional search engine is the ability to search for targeted information based on clearly defined information features. The main challenges of a targeted search engine are how to automatically find specific data in a highly unstructured environment, content-wise (i.e., a webpage), and how to efficiently process and store that data in order to make it accessible to the user.

Tackling all the possible scenarios and solutions in detail is an impossible task; therefore, several limitations are imposed on the considered environment. As previously mentioned, the focus is on extracting product information, but the location of the product features within the webpage is given for each website in the form of a configuration file. The method of obtaining that configuration file is beyond the scope of this paper, and it will be researched and properly tested by means of the proposed solution. Moreover, the novel method of increasing cost effectiveness when storing data on a cloud platform is better tailored to finding products rather than searching for other types of information.

To summarize, the considered scenario details and limitations are as follows:

1. The user can find product information pertaining to a single specific vertical.
2. The overall number of crawled websites is rather small and known beforehand.
3. The location in the webpage of the items that have to be extracted is determined manually for each website and is provided to the system in a configuration file.
4. The search engine must be publicly available to the end-user.

The current state of the proposed solution has the aforementioned limitations, but throughout this paper, various solutions to improve the system and to make it scalable are presented.

### 2.2. System Architecture

As mentioned previously, the research presented in this paper is based on the distributed framework from [8], where a highly modularized system architecture is presented without going into detail in regards to each component and without any optimizations or security considerations. Compared with the framework that was presented there, in this paper, the proposed solution streamlines the previous version and describes in detail the implementation specifics that are designed for cost effectiveness when deployed on a cloud platform.

An overview of the proposed system architecture is presented in Figure 1. The information flow is as follows: at a specified period in time (e.g., daily), a dispatcher starts a crawler for each website, and an extractor obtains the product information, which is then processed and deployed on a cloud web application in order to be accessed by the user.

The proposed solution can be summarized by four actions: retrieve, extract, process, and present. Each action represents a stage in the information flow, and each one can be easily replaced with a different implementation. For many of the identified challenges, the chosen solutions are basic in the sense that they represent the path of least resistance to obtain the goal of having a cost-effective targeted shopping search engine hosted on a cloud platform. In addition, the retrieval, extraction, and processing actions are performed on a local server, while the presentation action is performed on a cloud platform. This decision was taken due to the fact that running computing-intensive tasks on the cloud incurs a higher cost, rather than running them locally and just uploading the results to the cloud platform. Of course, this is feasible only in certain conditions, which are discussed later in this paper.

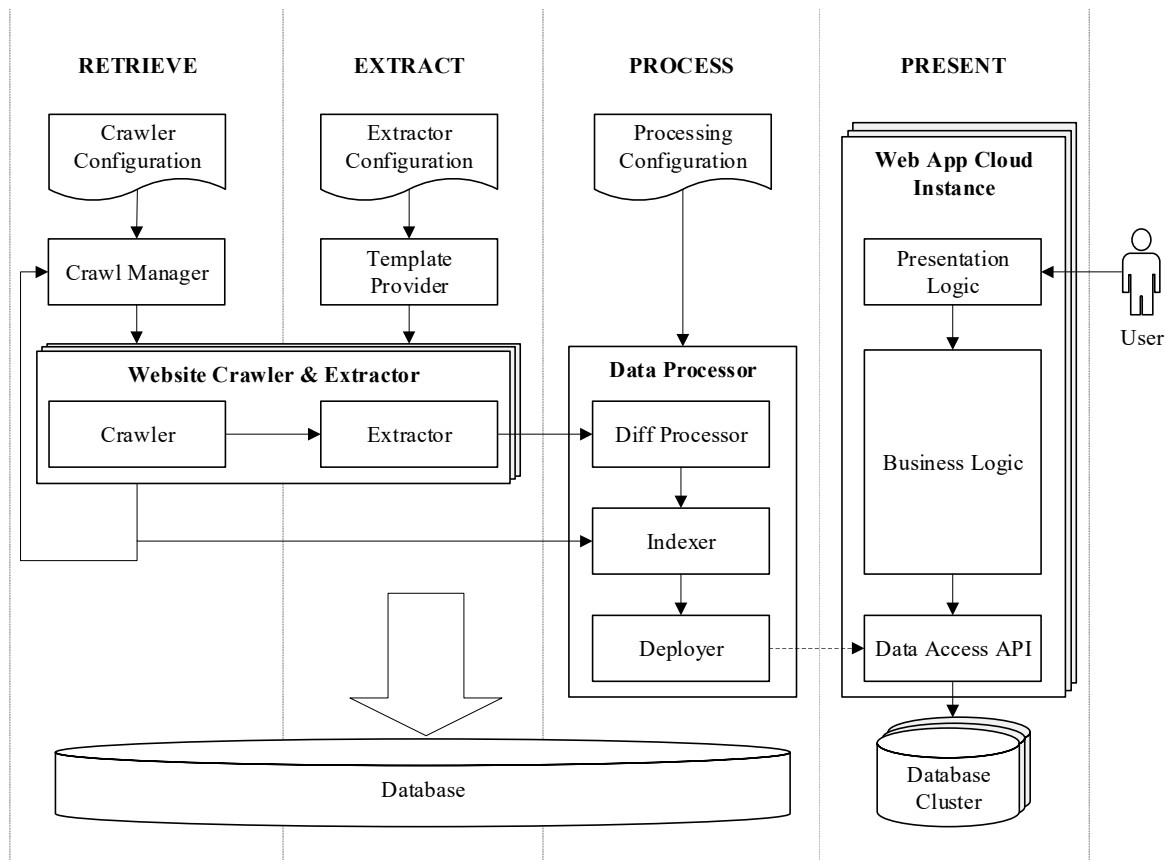

**Figure 1.** Overview of the proposed solution architecture and the information flow between the main components. For the first three actions (i.e., retrieve, extract, process) the components are on a local server and share the same database. For the last action (i.e., present), the components are deployed on a cloud platform.

### 2.3. Retrieve Action

The crawler needs a seed list of websites to crawl first and from which to extract all the links, which are added to the database and subsequently crawled. In theory, with the proper seed list, the crawler can go through all of the surface web; the deep web, for the purposes of this paper and product extraction, does not contain relevant product information [10]. For a vertical search engine, crawling most of the web is not an efficient solution because only an incredibly small percentage of the websites contain information relevant to that vertical. Instead of hectically scanning the web, the chosen trivial solution is to have a seed list with all the considered websites from that vertical and to process only those websites. In this situation, we have a crawler in the sense that it finds new webpages, but these are only from the initial website pool.

The retrieval process is coordinated by the crawl manager component, which has three subcomponents: the initializer, the scheduler, and the dispatcher. A configuration file is used to obtain the seed list and the crawling parameters; the former is used by the initializer to populate the database, while the latter are used by the scheduler. Keeping in mind that the goal is to have a cost-effective solution, the crawler will only access, if possible, pages that contain multiple product details. The idea is to crawl as few pages as necessary to extract the required information.

### 2.4. Proposed Method for Representing the Crawler Configuration

There are two parts to the crawler configuration. Firstly, the global crawl parameters that specify how the webpages are retrieved, and secondly, the site-specific parameters that describe what sites and what pages are going to be retrieved.

In terms of the global crawl parameters, these are:

− *database connection string*—host, port, username, password, database name, and type;
− *minimum and maximum waiting times* between crawls from the same site—a random value between that interval is chosen when retrieving each page;
− *recrawl interval/time*—the recrawl takes place after a specified time from the last recrawl or at a specified time each day;
− *web service URL and credentials*—used to upload the site and product information;
− *a list of site-specific parameters.*

For each site, the proposed system receives a list of parameters that are used by both the crawler, which needs to know what pages must be retrieved, and the extractor, which requires the location on the webpage of the product information that must be extracted. Therefore, the list of site-specific parameters is as follows:

− *name, url, logoUrl*—the site name, URL, and logo URL, which are needed by the presentation system component; the *url* property also serves as unique key for the purpose of determining if the site already exists in the database;
− *fetchUrlRegEx*—regular expression used to determine if a URL is added to the database for retrieval;
− *seedList*—list of webpages used to start the crawling process on that website;
− *extractor*—an object describing the extraction parameters:

    − *extractUrlRegEx*—a regular expression denoting which pages contain extractable data;
    − *baseSelector*—a custom selector describing the in-page location of the information that needs to be extracted;
    − *properties*—an object with key-value pairs, where the key is the product feature name, and the value is a custom selector relative to the base selector and contains the feature value that is to be extracted;
    − *uniqueKeyProperty*—property name from the properties object that serves as a unique key for the purpose of uniquely identifying a product belonging to a website.

The advantage of this representation is the flexibility that allows new websites to be easily added and allows different types of products to have completely different properties. Moreover, the uniqueKeyProperty offers the possibility of specifying the criteria for determining if a product is the same as one from the previous crawl (it can be the URL for some websites, or a product number for others).

*2.5. Extract Action*

After each a page is retrieved, for the sake of cost-effectiveness, it is sent to the extractor component, which obtains the product information. Another approach would have been to completely separate the crawler and the extractor, i.e., the crawler stores the page in the database, and a separate extractor process retrieves the page from the database and obtains the product data. The former solution was chosen to reduce the number of calls to the database and the volume size of the stored information at the expense of perhaps having a less scalable system. More details about this choice are presented in the Discussion section of this paper.

Extracting the information implies the knowledge of where the product features are on the page, i.e., a page template. At this point, the template is set manually for each website and is given as input to the extractor component in the form of a configuration file. A continuation of the research presented in this paper is to study the possibility of determining the template automatically for each website. There are multiple approaches to this problem, as shown in the review in [6].

Depending on what information is to be extracted, each website is analyzed so that all the products are obtained with the minimum number of webpages being retrieved. In accordance, multi-product

pages are favored as long as they contain all the required product features. In a majority of situations, only the product name, price, and availability are shown on those webpages, and for the extra features, the crawler needs to visit the single-product pages. The same product on different retail websites should have the same characteristics, which can be taken from the manufacturer website or from just one of the retailers. This means that, from the rest of retailers, we need only a means of getting the price and availability, after identifying that it is the same product. Determining that two products from different websites are the same is a difficult problem that will be researched as a continuation of the work from this paper.

*2.6. Proposed Method for Representing the Extractor Template*

The location of each product/item feature on the webpage varies on each website, and once it is determined, the extractor must be able to obtain the required data from the webpages that are identified as containing product information. An important challenge is how to represent the location of each product feature so that it can be quickly extracted from the webpage. This translates into reverse engineering the webpage template, or at least the part of the webpage that contains product information. Many existing crawler solutions extract data by means of CSS-like selectors or by using XPath, and then use library-specific methods to extract text or attributes from the selected HTML elements. Representing the product extraction template using only those methods is a problem that can be solved by adding extra functionality to the selector.

The new method for extracting data presented in this paper extends upon the JSOUP library [11], which allows the selection of HTML elements, and improves it by allowing the selection of attributes, text, and a Boolean value if a certain condition is met. A CSS selector is designed to identify only HTML elements; so in order to extract information, if the selector identifies an element, then all the text from that element (including all the text from its children) is returned by the proposed extractor. Besides all the CSS selector features, new symbols are added to allow more information to be extracted, as shown in Table 1.

**Table 1.** The custom selector symbols that were added and are used by the product information extractor.

| Symbol | Usage | Description |
|---|---|---|
| /<br>(slash) | selector/selector | Separates multiple selectors. The value that is returned is the one found by the first matching selector in the list. If the custom selector ends with a slash and no element was selected, then an empty string is returned instead of an exception being thrown. |
| %<br>(percent) | selector%attr<br><br>selector% | Returns the value for the attribute name following the symbol from the element selected by the string preceding the symbol.<br>If there is no expression after the symbol, then it returns the element text not including the text from that node's children elements. |
| =<br>(equals) | selector%attr=expr<br>selector%=expr | Returns a Boolean value depending on whether the left-hand value obtained from the selector matches the right-hand expression. |
| ~<br>(tilde) | selector%attr=~expr | Returns a Boolean by negating a right-hand expression when used with the equals sign. |
| abs:<br>(abs colon) | selector%abs:attr | Returns the absolute URL for an attribute value; can only be used after the percent symbol (%). |

Examples of custom selectors:

- `td.productlisting_price>.productSpecialPrice/td.productlisting_price>.price` returns the text from the element that has the *productSpecial* class attribute, which has a *<td>* parent element with class *productlisting_price*; if there is no such element, then it returns the text from the element with class *price,* which has the same parent.
- `div.product-details>p.price%` returns the text from element *<p>* with class *price* (not including the text from the child elements), which has a *<div>* parent with class *product-details.*

- `div.image>img%class=~outofstock` returns true if it is not *outofstock,* the value of the ** element's *class* attribute, which is a child of a *<div>* element with the class *image*.
- `div.name>a%abs:href` returns the absolute URL from the value of the *href* attribute of an *<a>* element having a parent of type *<div>* with class *name*.

The proposed approach to representing the extraction template is a flexible way of specifying the location of each product feature in a webpage. A direction for future research regarding this topic can be to determine a method of automatically obtaining the template, maybe using unsupervised learning techniques.

### *2.7. Process Action*

The data processor determines what information will be deployed to the presentation layer (i.e., the remote web server). There are two important types of information that are sent: the product data and the product index. Again, the goal is to minimize resource consumption at the presentation layer. This is why each extracted product data is sent to the diff processor, which determines if there are any changes compared to the previous crawl. Only the products that have different information are sent to the remote server. In order to minimize calls to the remote server's data access API, the products are sent in batches of 50 and each product in a batch is from the same website, as this can be used to optimize the data storing process at the server.

An important issue is how the presentation layer distinguishes between products that have no feature changes and products that are no longer on the website. Running the indexer at the process layer partly solves this problem by indexing only the products that are extracted at the last crawl. This means that when the user performs a search, only the products that are currently present on the websites are returned as results. Unfortunately, this poses a dilemma regarding the products that are no longer on the websites: should they appear in the search results or not? Usually, when a product is removed from a retailer website, it is removed from that website's search and the product page is not accessible by navigating the website; however, it might be accessible by using an external search engine. If we also take into account that a shopping search engine needs to have a price history component, the chosen option is to leave the products in the database and to just mark them as permanently out of stock.

The indexer's role is to split each product name into words and to obtain for each word a list of products whose names contains that word; this is a typical example of the MapReduce algorithm [12]. In its current state, the indexer allows only words that have a length greater than three and contain no prepositions. This simple approach is enough for the proof-of-concept.

### *2.8. Present Action*

This action is represented by a web application deployed on a cloud instance, which allows the end-user to search products by name. Other main features of the presentation solution are: the ability to view the list of products that were added in the last week, to view the list of retailer websites that were crawled, and to save the desired products in the user's favorites list. Another useful feature is the product price history chart, which allows the user to see the evolution of the product price. There are many features that can be implemented, but the main focus is to minimize resource usage, especially at the presentation layer.

Regarding the communication between the processing and the presentation layers of the proposed solution, this is achieved by means of RESTful micro web services, as shown in Table 2. The communication situations are designed to have a low impact on the number of requests and on the number of reads and writes on the database.

**Table 2.** Exposed RESTful micro web services at the presentation layer.

| URL | HTTP Method | Payload / Query | Description |
|---|---|---|---|
| /sites | GET | - | Returns the websites |
| | POST [1] | *website* | Adds a new website |
| | PUT [1] | *website list* | Replaces the website list |
| | DELETE [1] | - | Removes all the websites |
| /sites/*siteUrl* | PUT [1] | *website* | Replaces the website |
| /products | GET | search=*terms* | Performs a product search by name |
| | | newest=true | Returns the products that were added in the last week (limited to 100 items) |
| | | ids=*listOfIds* | Returns the products with the specified IDs (limited to 100 items) |
| | PATCH [1] | *product list* | Batch-updates the products |
| | DELETE [1] | - | Removes all the products |
| /productindex | GET [1] | | Returns the product index |
| | PUT [1] | *product index* | Replaces the product index |

[1] Method available to be called only from the processing layer.

Optimizations are made on the presentation server by using a memory cache, which is shared among the cloud-created instances of the web application, thus reducing the communication with the database. In order to lower costs, the product index is kept in the database and in the memory, rather than relying on the database's indexing.

*2.9. Proposed Solution for Product Indexing*

When the user enters a search string, for each word entered, a lookup is performed in the product index. Afterwards, if there is more than one word in the search string, an intersection of the results for each word is performed. The product index is a Patricia Trie data structure, which can be distributed as shown in [13], where the key is the word and the associated value is a list of product IDs. For each ID, the product information is retrieved from the memory cache with the database as fallback. In terms of time complexity in the average case, the product indexing is performed in $O(n)$, where $n$ is the number of products that are indexed, while the product search is performed in $O(1)$. These low time complexities are possible because a Patricia trie is used to store the information.

The separate indexing approach has the advantage of allowing optimizations to be made regarding the database access, i.e., reducing the number of reads and writes. Currently, the product index is compressed and is kept as an entry in the cloud database, where the key is an empty string. If the entry gets too large, the storage method takes advantage of the trie structure and splits the index by the string prefix. Further research regarding the indexing will pe performed in order to achieve better scalability.

An advantage of not relying on the database indexing solution is that database access costs are reduced, at the expense of some computational costs. Moreover, the computation of the index can be performed either on the crawler, extractor, and processor side, or on the deployment side. Taking a low-level approach and moving the indexing to the cloud can benefit from the cloud's push/pull queues feature. On the other hand, the indexing logic can be easily extended to use third-party solutions like Apache Lucerne. Currently, the indexing is performed locally as the product information is processed, but there is also a re-indexing service implemented on the cloud application.

*2.10. Security and Ethics*

2.10.1. Crawler Security and Ethics

The proposed crawling solution simulates the behavior of a user by sending requests at random times and by sending the appropriate HTTP headers to the crawled server as if the request was made from a browser.

In terms of ethics, a crawler must adhere to the robots exclusion standard by processing a robots.txt file, which regulates what pages can and cannot be visited by a crawler on a specific website [14]. If those rules are not obeyed by the crawler, there is a slight chance of getting itself trapped in a honeypot [15]. For example, in a webpage there can be links to fake webpages that are not visible in the browser, which, when crawled, result in fake product data or in other fake webpages, maybe from different websites.

Usually, retail websites want to have their products indexed by a search engine, but the crawler must be designed to prevent situations in which it retrieves and processes useless webpages. From this point of view, the proposed solution ensures the crawler's security by specifying, in the website template and with the help of regular expressions, what URLs are crawled and what URLs are extracted.

### 2.10.2. Presentation Layer Security

The presentation layer hosted in the cloud exposes RESTful web services in order for the product information to be added to the database and allows the users to search for products by name. Regarding the security issues identified in [16,17], the cloud solution where the presentation layer is deployed ensures server security, and in terms of the web application, the appropriate measures have been taken to ensure protection from attackers taking advantage.

At this stage, there is no possibility of end-users logging in on the website, so there is no user access and control to secure. However, there is the communication between the processing and the presentation layer, which is protected from man-in-the-middle attacks through HTTPS. In addition, some of the micro web services can only be accessed by the deployer component of the processing layer via IP filtering. Protection against improper input validation (e.g., SQL injections, cross-site scripting) is achieved by allowing only letters and numbers in the website's search input and by limiting the number of characters in the search string. The end-user is able to call only two API methods: to get all the site information and to get product information, as can be seen in Table 2. Regarding the favorite products logic, this is performed in the browser using the local storage and has no influence on the website's security. As the presentation component evolves, penetration testing tools will be employed to prevent attackers from exploiting the application.

Another aspect is third-party applications using the data presented on the proposed website. In a sense, this is similar to having a search engine and preventing other applications from using the search results. The first employed step is the cross-origin resource sharing (CORS) method, which controls access when requests are coming from other origins. Unfortunately, this does not prevent another web application to simply do a GET request server-side and to obtain the data. The best way to protect from third-party applications is to detect that the request came from an application rather than an end-user. This situation is similar to the one discussed in the Crawler Security subsection of this paper, but this time the presentation layer is the victim, whereas there, the crawled websites were the victims and the crawler was the attacker.

## 3. Use—Case Scenario and Results

The goal is to have a fully-functional vertical search engine, and in order to test the proposed solution, the chosen vertical is board and card games. Seven websites are crawled: barlogulcujocuri.ro, lelegames.ro, lexshop.ro, ludicus.ro, pionul.ro, redgoblin.ro, and regatuljocurilor.ro. The deployed website for the end-user is shown in Figure 2 and is accessible at: http://boardgamesearch.h23.ro/.

All of the crawling, extraction, and processing is performed on the same computer for an accurate measurement of the system performance. There are two main applications hosting the solution components. The first is a Java application that uses a MongoDB database and hosts the crawling, extraction, and processing components. For the presentation component, a Java-based web application is used and deployed on the Google Cloud platform with Google Datastore as the database [18]. Other employed technologies are Jersey RESTful web services, memory cache services, and push queues.

The reasoning behind choosing the Google Cloud platform is that it is free with daily limitations; every 24 h the quotas are reset.

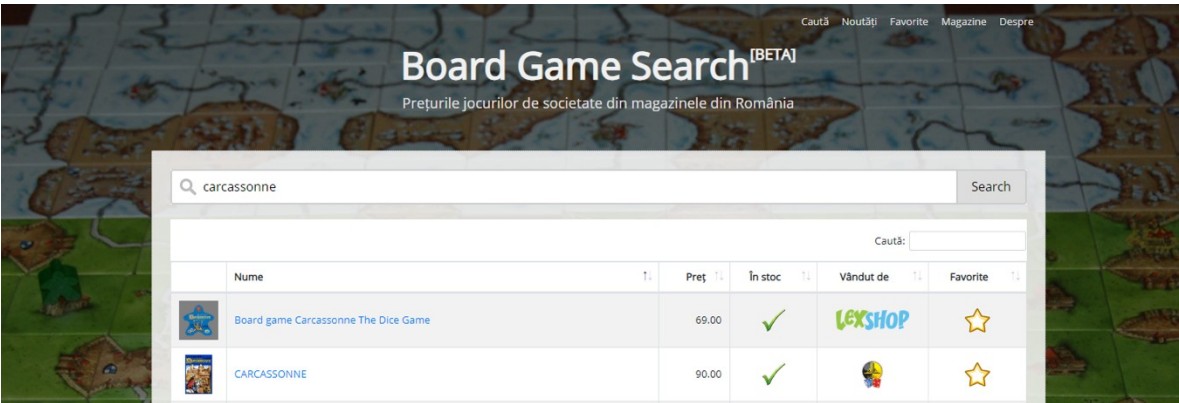

**Figure 2.** Results page for the board game search website: http://boardgamesearch.h23.ro.

For each website, a crawler and extractor configuration file are used, as can be seen in Listing 1 for the lexshop.ro website. For each product, the properties/features that are extracted are: URL, name, image, price, and availability, which represent the bare minimum for offering relevant results to the user. Ideally, the configuration is to be obtained manually, but for now, as previously mentioned, it is entered manually after analyzing the websites in question. More aspects regarding the website template analysis are presented in the Discussion section of this paper.

```
{
    "name": "lexshop.ro",
    "url": "https://www.lexshop.ro",
    "logoUrl": "https://www.lexshop.ro/app/images/logo.png",
    "fetchUrlRegEx": "\\Qhttps://www.lexshop.ro/?page=produse&categorie=8&n=\\E[0-9]*",
    "seedList": [
"c8-board-games"
    ],
    "extractor": {
            "extractUrlRegEx": "\\Qhttps://www.lexshop.ro/?page=produse&categorie=8&n=
            \\E[0-9]*",
            "baseSelector": "div.list-products>div>div>div[data-href]",
            "uniqueKeyProperty": "url",
            "properties": {
                    "url": "div.product_img_container>a%abs:href",
                    "name": "div.prod_title_container>a",
                    "image": "div.product_img_container>a>img%abs:data-original",
                    "price": "div.prod_prices>span.actual_price",
                    "availability": "div.product_img_container>div.eticheta-stoc=
                    ~STOC EPUIZAT"
            }
    }
}
```

Listing 1. Sample crawler configuration and extractor template file representing a single website.

The crawl, extraction and processing times per page for each of seven online board game stores are shown in Table A1 (in Appendix A), where the processing time is the sum of the page retrieval, link extraction, product extraction, and server upload times. The sum values from the table depend on the number of links found and the number of extracted products from each page; these values are presented in Table A2 (in Appendix A), The volume of data that is extracted from each website varies significantly. For example, on one website there are around 852 links per page, while on another there are 190 links. In addition, some sites permit the product display with a maximum pagination of 60, while others only allow 12 products per page. All of this leads to different page processing times. Overall, from the 851 retrieved pages, there were 14860 unique products extracted in 1937 s. There were approximately 17 products per page, and the time to process a product in these conditions is around 0.13 s. The compressed product index is kept in the datastore in a blob of 275 KiB; the size is significantly affected by the fact that, for now, the product IDs are strings instead of long-type values.

An interesting aspect is that for some sites, the number of unique products found in the last crawl is significantly less than the number of unique products stored in the database. For example, the Red Goblin website currently has 4575 products in the database, but in the last run, only 2037 products got extracted. This is due to the fact that 2538 products were previously available in the last year on the site but have been removed since then. Four of the six websites were crawled for approximately one year, and two among them have removed most of the products that are out of stock, while the other two just change their availability.

## 4. Discussion

The system architecture decisions are presented in Section 2 together with a description of the components. Here, there is a focus on the implications that the results might have on expanding and improving the proposed solution.

In order to minimize the number of pages that are processed, the information is extracted from multi-product pages. From a single webpage, on average, there are 17 unique products extracted. The advantage is obvious: less requests to the servers hosting the websites, less chance to ban the crawler for initiating too many requests, and most importantly, less time to extract the information. This works because we need only the bare minimum of information. On the other hand, for example, if one wants to also extract the product description, then the regular expression that matches the page URLs used for extraction will have to be changed to match single-product pages that have that description. Moreover, all of the custom selectors will have to be updated to the single-product page template. Figure 3 shows the number of extracted products per second from each considered retailer. These values depend mostly on the webpage content size and secondly, on the number of extracted pages and the number of products per page. For example, on the Pionul website, an average of 14.57 products/page were extracted, compared to 1.96 products/page on the Barlogul cu jocuri website, because the latter had a smaller average page content size. Considering all of the page processing times, it takes the longest time to retrieve the webpage, as can be seen in Figure A1 (in Appendix B), which shows the total sum of the processing times in relation to the total number of uniquely extracted products for each of the seven considered websites.

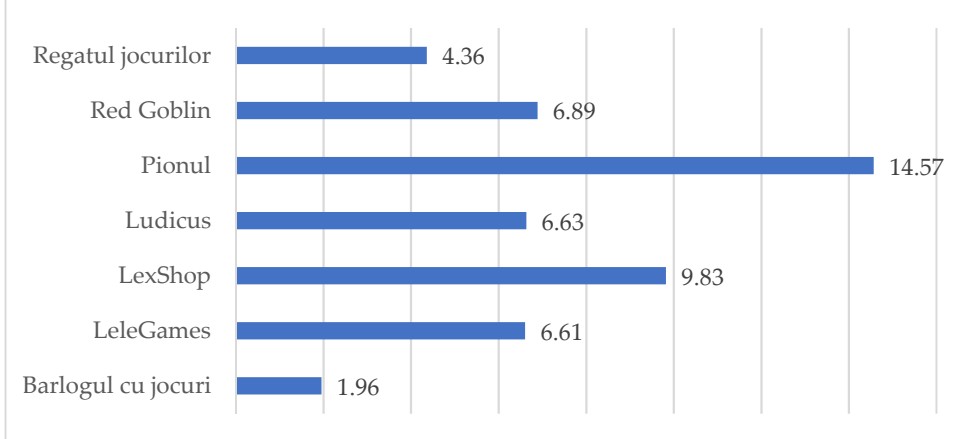

**Figure 3.** Number of extracted products per second for each of the seven considered retail websites.

A site that was in contention to be crawled was transylvaniagames.com. The problem with the site is that the product availability is not present in the multi-product pages and, more importantly, the product URLs are in the form transylvaniagames.com/*product-name*. This poses the problem of distinguishing between product pages and the other pages. The best solution is to go from the multi-product pages to the product pages and extract from there. Therefore, the next step is to extend the template to allow, in some cases, the retrieval and processing of the extracted product URLs, which are always present on multi-product webpages.

Another issue is what is considered to be the unique key, which is used to determine if a product that was extracted is the same as a product that is already in the database. In practice, and considering that there is no product that can be accessible by two different URLs, the unique key is composed of the site ID and the product page URL path: it needs to be composed because we can have the same URL path on different websites.

For all seven websites, the page retrieval time is approximately ten times the sum of the link extraction, product extraction, and server upload times. This means that if we want to achieve scalability, it would make sense to have more instances that just retrieve the webpage and store it in a database, compared to the number of instances that perform the extraction and upload. The problem with storing the webpage content in the database is in terms of storage and the transfer times to store and to retrieve the page content. An optimization for storage is to compress the page before saving it in the database. Another advantage is that the crawler can use the cache-related HTTP headers to determine if the webpage has been modified since the last crawl. Unfortunately, in most situations the cache-control header is set to no-cache, or it is not accurate. From a crawler's point of view, the best way is to use a pool of crawlers, with each crawler obtaining from the database the URL that was accessed earliest and comes from a website that wasn't crawled in the last seconds. In this way, we have no crawler downtime and the website servers do not ban the crawler for making too many requests.

In terms of resource usage for a shopping search engine, there are a few key aspects that need to be optimized: the incoming bandwidth, the instance hours, and the datastore read and write operations. By having the presentation component on the cloud, we are left to optimize only the datastore read and write operations. In the free version of the Google Cloud platform, these are limited to 50,000 reads and 20,000 writes. The writes are consumed when deploying products on the remote web server, and the reads are consumed when the user performs a search, or visits the latest products section or the favorites section. While the cache service helps to significantly reduce the number of database reads, if no one accesses the website for a certain amount of time, the cloud instance of the website is removed from memory together with the cache. Initially, each product was stored as one entry in the database, but a more efficient way is to store multiple products in the same entry. This is feasible because the proposed solution keeps its own product index. We can obtain another increase in efficiency if we can

minimize the number of database entries that are retrieved when a search is performed. Thusly, it will help to keep track of the most searched terms and keep most of the results in a single entry in the database. The drawback is that there is a limitation on the entry size of approximately 1 MiB. This means that we need a function to determine if the entry is less than 1 MiB by computing the size in bytes of each product.

In terms of related work, existing shopping search engines rely on a product feed from the online retailers, which is usually in the form of an XML containing the elements: manufacturer, product name, category, product URL, price, identifier, image URLs, description, and other similar fields. Examples of shopping search engines that obtain product information from retailer feeds are: pricegrabber.com, shopzilla.com, shopping.com, compare.ro, and price.ro.

- *Similarities between shopping search engines that use retailer-provided feeds and the proposed solution*

The end-result is the same: provide the user the means to search for products and find the best available price. In both situations, the search engine needs to periodically check if new products were added, if products were removed, and if the product information somehow changed. The traditional shopping search engines offer each retailer the possibility of specifying the feed update frequency, while the proposed solution crawls each website daily. On the other hand, the crawl frequency can differ for each website based on the product update history, e.g., some retailers add new products only at the beginning of the week, and other retailers rarely change the product prices.

- *Advantages of shopping search engines that use retailer-provided feeds vs. the proposed solution*

For existing shopping search engines, the product information is accurate, regardless of any change in the webpage's template; however, websites rarely change their page structure and the proposed solution sends a notification for a template update if a structure change is detected. There is little resource usage on the search engine server because the only processing is to interpret the XML feed for each website. The feeds use pull technology, i.e., the initial request in made by the client (search engine), and therefore any change in the product information is obtained only when the search engine performs a request to that retailer's feed.

- *Advantages of the proposed solution vs. shopping search engines that use retailer-provided feeds*

A critical advantage of the proposed solution is that it does not require any involvement on the retailer's part. In order to obtain a product feed, there needs to be an interaction between the administrators of the search engine and the retail website, and more often than not, the retailer has to pay a fee to have its products indexed. The proposed solution eliminates this interaction by simply crawling the retailer webpages and extracting the product information, even from websites that are not willing to provide a product feed. Another important advantage is that the proposed solution is designed to be generic and works on any type of extractable information (e.g., food recipes), compared to shopping search engines, which allow only product searches.

In regards to comparing the performance of the proposed solution with existing shopping search solutions, it is significantly difficult to compare resource usage because the third-party solutions do not provide any information of this type. On the other hand, there is one aspect that can be evaluated: the volume and accuracy of the product data. In this regard, the proposed solution has the advantage of having product information from more websites, not just the ones that want to provide XML feeds, while maintaining the same data accuracy. For board games, an example is the website boardgameshub.ro, which is a search engine that uses XML feeds but does not index products from big retailers that also sell other types of products (e.g., elefant.ro, carturesti.ro), because those retailers feel that they do not get any added benefit from providing such a feed to the board game search engine.

Another somewhat related solution in terms of product extraction is the web scraper from webscraper.io. It is a browser extension that allows the user to specify, on a webpage, the location

of items to be extracted, which are then saved in CSV format. The main disadvantage compared to the proposed solution is that the web scraper allows only simple extraction from HTML elements using CSS selectors, i.e., it does not permit conditional extraction. For example, there are websites that have some products on sale, and the real price has a different CSS selector compared to the normal products. The web scraper has the advantage of its simplicity and ease-of-use, but it only provides simple scraping, which is only a part of the solution proposed in this paper.

In terms of security, the main aspects are discussed in the Security and Ethics section of this paper. Regarding the interaction between the user and the board game search website, that interaction is minimal, i.e., the user can input a search string, which is sanitized in order to prevent cross-site scripting. The retrieval solution adheres to the crawler ethics, and therefore no website banned (even temporarily) the crawler component. In addition, the communication between the processing and the presentation component is not visible to the public, and the data transfer is secured by means of HTTPS.

## 5. Conclusions

The proposed solution is a fully-functional vertical search engine that works not only on products but also on other types of extractable information (e.g., food recipes). A novel contribution that this paper presents is a product retrieval solution that does not depend on XML feeds provided by the retailers, but rather provides a concise, flexible, and efficient method of retrieving and extracting information by employing a novel template used to represent the location of the extractable data from webpages. Another novel aspect is the cost-effective method of storing data on a cloud platform and indexing it to minimize resource usage, while also providing efficient solutions for reducing resource consumption and, therefore, the costs in the other system components. Finally, a discussion is presented on the ethics and security that the proposed system poses.

In terms of extending the research in the considered field, the proposed solution has the great potential of being a framework for developing and testing new methods for product extraction (e.g., determining the extraction template using neural networks or determining if a product is the same as one from another website even though it has a slightly different name). The next research step is to move the crawler onto the cloud platform and make it as efficient as possible. The indexing component is already moved to the cloud and uses Google's push queues to execute the tasks that perform the indexing. There are two types of indexing: a live indexing as the web server receives batches of products, and a re-indexing service, which splits the process into multiple tasks in order to follow the cloud's task duration restrictions. Another considered extension of the proposed solution is to deploy it on an OpenStack cloud platform.

**Funding:** This research received no external funding.

**Conflicts of Interest:** The authors declare no conflict of interest.

## Appendix A

**Table A1.** Crawl, extraction, and processing times per page for each of seven online board game stores (expressed in seconds). The processing time (written in bold) is the sum of the page retrieval, link extraction, product extraction, and server upload times.

|  | Barlogul cu jocuri | LeleGames | LexShop | Ludicus | Pionul | Red Goblin | Regatul jocurilor | Total |
|---|---|---|---|---|---|---|---|---|
| Sum of page retrieval times | 43.05 | 43.90 | 185.60 | 106.81 | 27.38 | 211.78 | 947.70 | 1566.21 |
| Sum of link extraction times | 0.30 | 2.13 | 8.94 | 1.00 | 4.22 | 5.25 | 19.39 | 41.22 |
| Sum of product extraction times | 2.75 | 9.04 | 32.42 | 23.09 | 10.44 | 54.30 | 51.08 | 183.11 |
| Sum of server upload times | 0.44 | 4.84 | 47.50 | 10.63 | 7.31 | 24.19 | 52.24 | 147.14 |
| **Sum of processing times** | **46.53** | **59.91** | **274.46** | **141.53** | **49.34** | **295.51** | **1070.41** | **1937.68** |
| Average of page retrieval times | 4.78 | 1.83 | 0.78 | 5.09 | 0.76 | 2.75 | 2.13 | 1.84 |
| Average of link extraction times | 0.03 | 0.09 | 0.04 | 0.05 | 0.12 | 0.07 | 0.04 | 0.05 |
| Average of product extraction times | 0.31 | 0.38 | 0.14 | 1.10 | 0.29 | 0.71 | 0.11 | 0.22 |
| Average of server upload times | 0.05 | 0.20 | 0.20 | 0.51 | 0.20 | 0.31 | 0.12 | 0.17 |
| **Average of processing times** | **5.17** | **2.50** | **1.15** | **6.74** | **1.37** | **3.84** | **2.41** | **2.28** |

**Table A2.** Number of links and products found per page, the number of retrieved pages, and the total number of uniquely extracted products in the last crawl.

|  | Barlogul cu jocuri | LeleGames | LexShop | Ludicus | Pionul | Red Goblin | Regatul jocurilor | Total |
|---|---|---|---|---|---|---|---|---|
| Sum of number of found links | 2065 | 4552 | 70682 | 5317 | 4858 | 65596 | 131527 | 284597 |
| Sum of number of found products | 103 | 432 | 2858 | 1246 | 719 | 4620 | 9151 | 19130 |
| Average of number of found links | 229 | 190 | 296 | 253 | 135 | 852 | 296 | 334 |
| Average of number of found products | 11 | 18 | 12 | 59 | 20 | 60 | 21 | 22 |
| Number of retrieved pages | 9 | 24 | 239 | 21 | 36 | 77 | 445 | 851 |
| Number of uniquely extracted products | 91 | 396 | 2697 | 939 | 719 | 2037 | 4672 | 11551 |

**Appendix B**

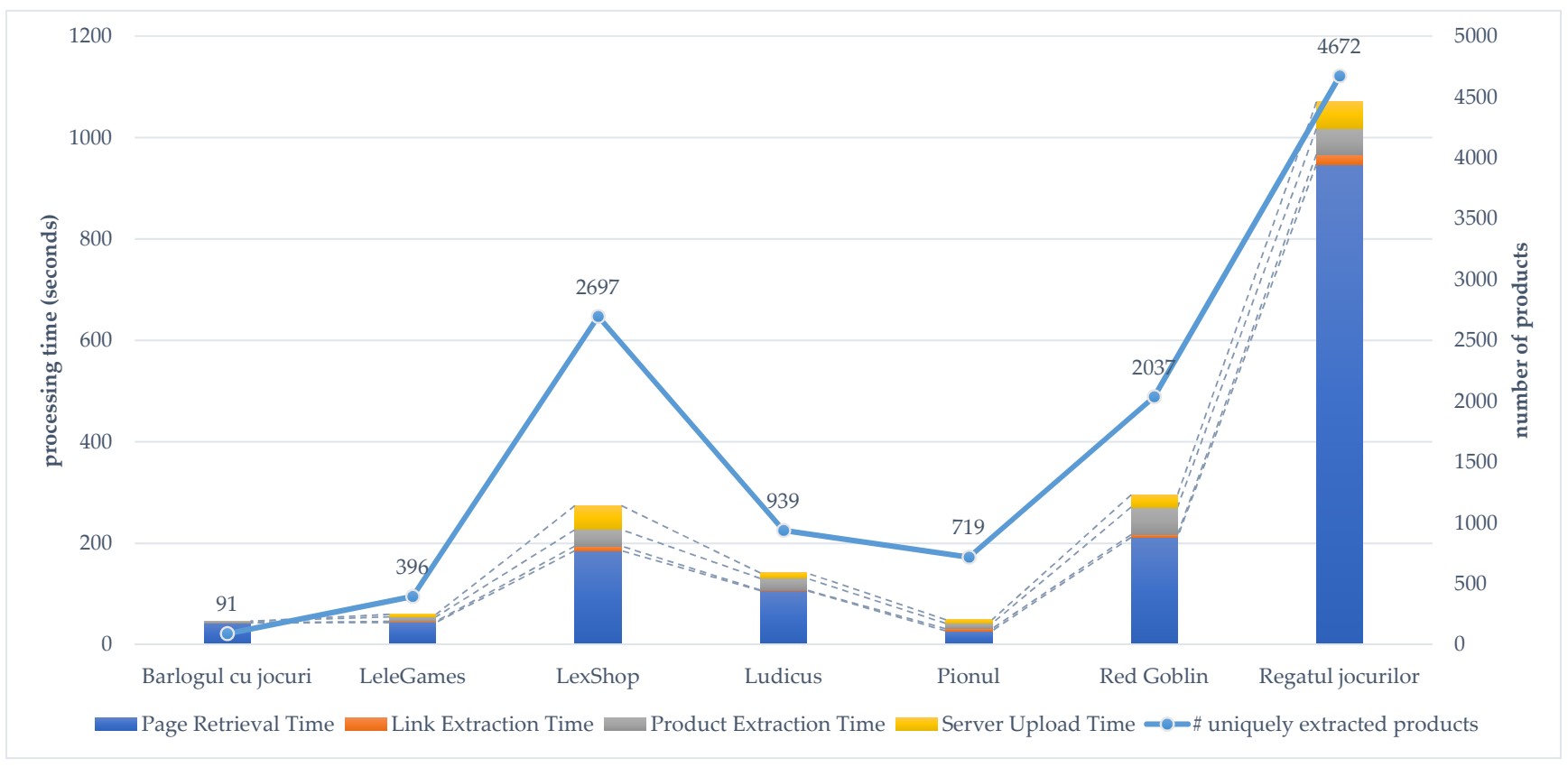

**Figure A1.** Total sum of processing times (i.e., page retrieval, link extraction, product extraction, and server upload times) in relation to the total number of extracted products from each of the seven considered websites.

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
