# Peer review of "Optimization and Security in Information Retrieval, Extraction, Processing, and Presentation on a Cloud Platform"

_information, doi:10.3390/info10060200_

Round 1

Reviewer 1 Report

The manuscript describes a vertical search engine, which exploits novel methods for crawling and extracting information, for product index optimizations and for deploying and storing data in the cloud database.

The overall engine is well described, together with the various steps which it follows to crawl, analyze, extract and represent information from web sites. The authors also briefly address security and privacy issues which may arise in the process.

Experimental results are presented, though there is no actual comparison with other search engines

Author Response

Point 1: Experimental results are presented, though there is no actual comparison with other search engines

Response 1: In the revised version of the paper, the Discussion section was significantly extended to include a comparison to other search engines and existing product scrapping methods.

Reviewer 2 Report

Authors proposed a method to represent the crawl and extraction template, for product index optimizations and for deploying and storing data in the cloud database. In addition, they discussed key aspects regarding ethics and security in the proposed solution. A practical use-case scenario is also presented, where products are extracted from seven board and card games online retailers.

There are some comments that can be applied to improve paper quality as follows:

1.      The paper must be proofread.

2.      As authors stated in the abstract, they discussed the security of the proposed method, but in the discussion section this section is not explained clearly.

3.      The discussion section can be improved by showing some graphs and tables.

4.      The proposed method must be compared to the most related works.

Author Response

Point 1: The paper must be proofread.

Response 1: The revised version of the paper was proofread by a CELTA-certified English teacher and legal translator.

Point 2: As authors stated in the abstract, they discussed the security of the proposed method, but in the discussion section this section is not explained clearly.

Response 2: There is section 2.10. Security and ethics, where these aspects are discussed, and, also, an overview in terms of security pertaining to the board game search use-case was added to the Discussion section.

Point 3: The discussion section can be improved by showing some graphs and tables.

Response 3: Besides the existing table in the Appendix, two more charts were added and discussed (Figures 3 and B1).

Point 4: The proposed method must be compared to the most related works.

Response 4: There is a brief mention of the related work in the Introduction section, but the Discussion section was significantly extended in the revised version to include a comparison between the proposed method and existing solutions.

Reviewer 3 Report

The author in this paper has presented a method for the processing steps to produce efficient vertical search engine including retrieval, extraction, presentation, and delivery. The paper has serious flaws and the following comments require serious attention:

1. What is the paper contribution? The current proposal does not count as a contribution. Many search engines are out there with excellent efficiency and performance. If I do not see a novel contribution, I can not qualify this as a work to be published!

2. The paper continues no section for literature study. Add a full section on related work. Consider work related to security and privacy, AI and deep learning. I recommend the following paper to enrich your article:

- M. Aloqaily, I. Al Ridhawi, H. B. Salameh, Y. Jararweh, Data and service management in densely crowded environments: Challenges, opportunities, and recent developments, IEEE Communications Magazine.

-  S. Otoum, B. Kantarci and H. T. Mouftah, "On the Feasibility of Deep Learning in Sensor Network Intrusion Detection," in IEEE Networking Letters.

Silva R., Iqbal R.: “Ethical Implications of Social Internet of Vehicle Systems”, IEEE Internet of Things Journal, Early Access, 2018.

Kendrick, Phillip, et al. "An Efficient Multi-Cloud Service Composition Using a Distributed Multiagent-based, Memory-driven Approach." IEEE Transactions on Sustainable Computing (2018).

- Aloqaily, Moayad, et al. "An Intrusion Detection System for Connected Vehicles in Smart Cities." Ad Hoc Networks (2019).

- S. Otoum, B. Kantarci, and H. Mouftah, "Adaptively supervised and intrusion-aware data aggregation for wireless sensor clusters in critical infrastructures " 2018 IEEE International Conference on Communications (ICC), Kansas City, MO, 2018, pp. 1-6.- Oueida, Soraia, et al. "An Edge Computing Based Smart Healthcare Framework for Resource Management." Sensors 18.12 (2018): 4307.

- D. Turgut and L. Boloni. Value of Information and Cost of Privacy in the Internet of Things. IEEE Communications Magazine, 55(9):62ذ66, September 2017.

- S. Otoum, B. Kantarci and H. T. Mouftah, "Detection of Known and Unknown Intrusive Sensor Behavior in Critical Applications," in IEEE Sensors Letters, vol. 1, no. 5, pp. 1-4, Oct. 2017, Art no. 7500804.

3. Your paper title includes an Optimization keyword. However, I have seen where exactly the optimization has been done or apploed to?

4. I have not seen any results of comparison at the performance level?

5. What is the level of complexity of the proposed algorithm?

6. The conclusion section is written in the non-academic stander. 

Author Response

Point 1: What is the paper contribution? The current proposal does not count as a contribution. Many search engines are out there with excellent efficiency and performance. If I do not see a novel contribution, I can not qualify this as a work to be published!

Response 1: The novel contribution is clearly stated through the paper, and in concise form in the Conclusions section of the paper. For example, the existing search engines rely on product feeds from the retailers, which are not needed by the proposed solution. The Discussion section was extended and the Conclusions section was rewritten to better emphasize the novelty of the paper.

Point 2: The paper continues no section for literature study. Add a full section on related work. Consider work related to security and privacy, AI and deep learning. I recommend the following paper to enrich your article:

1)      M. Aloqaily, I. Al Ridhawi, H. B. Salameh, Y. Jararweh, Data and service management in densely crowded environments: Challenges, opportunities, and recent developments, IEEE Communications Magazine.

2)      S. Otoum, B. Kantarci and H. T. Mouftah, "On the Feasibility of Deep Learning in Sensor Network Intrusion Detection," in IEEE Networking Letters.

3)      Silva R., Iqbal R.: “Ethical Implications of Social Internet of Vehicle Systems”, IEEE Internet of Things Journal, Early Access, 2018.

4)      Kendrick, Phillip, et al. "An Efficient Multi-Cloud Service Composition Using a Distributed Multiagent-based, Memory-driven Approach." IEEE Transactions on Sustainable Computing (2018).

5)      Aloqaily, Moayad, et al. "An Intrusion Detection System for Connected Vehicles in Smart Cities." Ad Hoc Networks (2019).

6)      S. Otoum, B. Kantarci, and H. Mouftah, "Adaptively supervised and intrusion-aware data aggregation for wireless sensor clusters in critical infrastructures " 2018 IEEE International Conference on Communications (ICC), Kansas City, MO, 2018, pp. 1-6.- Oueida, Soraia, et al. "An Edge Computing Based Smart Healthcare Framework for Resource Management." Sensors 18.12 (2018): 4307.

7)      D. Turgut and L. Boloni. Value of Information and Cost of Privacy in the Internet of Things. IEEE Communications Magazine, 55(9):62ذ66, September 2017.

8)      S. Otoum, B. Kantarci and H. T. Mouftah, "Detection of Known and Unknown Intrusive Sensor Behavior in Critical Applications," in IEEE Sensors Letters, vol. 1, no. 5, pp. 1-4, Oct. 2017, Art no. 7500804.

Response 2: Following the paper template guidelines, the related work section is part of the introduction. On the other hand, in the revised version of the paper, the Discussion section was significantly extended to include comparisons with similar existing solutions. In terms of the reviewer’s recommendations, after reading all the paper abstracts and some of the papers in full, the conclusion is that those papers have no or insignificant relevance to the proposed solution, which is about having an efficient vertical search engine.

Nonetheless, here are the justifications for each suggested paper:

1)      That paper discusses data replication and service composition in densely crowded environments (stadiums, metro stations), which has nothing to do with search engines.

2)      That paper deals with intrusion detection in wireless sensor networks, which also has nothing to do with search engine security.

3)      Even from the title, one can clearly see that there is not common point between that paper and any of the concepts discussed in our paper.

4)      Compared to the other suggestions, that one is more to the point because it deals with reducing the energy and the price cost of fulfilling requests in distributed cloud services. Nonetheless, it does not warrant a reference, because the proposed solution uses cloud resources and it does not deal with distribution across different data centers.

5)      Same remark as for the third suggested paper.

6)      Wireless sensor clusters have nothing to do with search engines.

7)      That paper has some interesting ideas regarding privacy, but it deals with the Internet of Things, and, again, does not warrant a reference.

8)      Again, wireless sensor networks have nothing to do with search engines.

Point 3: Your paper title includes an Optimization keyword. However, I have seen where exactly the optimization has been done or apploed to?

Response 3: The optimizations are applied to the whole information flow in order to increase the processing speed and the Cloud resource cost. Some paragraphs have been modified to better emphasize this.

Point 4: I have not seen any results of comparison at the performance level?

Response 4: The performance comparison between the processing of each considered website is discussed throughout the Use-Case Scenario and Results section and the Discussion section of the paper. In terms of the performance comparison between the proposed solution and existing shopping search engines, in the revised version of the paper, the Discussion section was extended to address this issue.

Point 5: What is the level of complexity of the proposed algorithm?

Response 5: The proposed solution is not a single algorithm per se, but rather an architecture with many algorithms (e.g., crawling, link and product extraction, indexing, product search). The complexity discussion for the product indexing and search was added, but the complexities of for the other proposed methods are not that easily quantifiable. Nonetheless, short discussions in terms of effectiveness can be found when the system components are presented.

Point 6: The conclusion section is written in the non-academic stander.

Response 6: In the revised version of the paper the Conclusions section was rewritten to adhere to the academic standard.

Round 2

Reviewer 3 Report

I agree with your justifications, however, I still have the following comments:

The works presented with 2, 6, and 8 is pure algorithms on security in information communication retrieval, extraction, processing.

I am still not convinced how fundamental this software is? Let me help with this. What the novel contribution of this software for someone or for a corporate to buy or invest in? What are its benefits and how these benefits are not already out there?

What is your benchmark test for the optimization? Can you show us some Figures?

Author Response

Point 1: I agree with your justifications, however, I still have the following comments. The works presented with 2, 6, and 8 is pure algorithms on security in information communication retrieval, extraction, processing.

Response 1: I have read again all three suggested papers in their entirety. The algorithms that are tested there are applicable in sensor networks. They have nothing to do with retrieving and extracting product information from the web, processing that information and presenting it to the user. Again, securing a sensor network is significantly different than securing a product search engine solution.

It would have been more relevant to reference “Computer Security Fundamentals, 3rd Edition” by William (Chuck) Easttom, or “Information Security Fundamentals, 2nd Edition” by Thomas Peltier, but those books contain concepts that are generally known in the field.

The security implications are presented in section “2.10. Security and ethics” of the submitted paper.

Point 2: I am still not convinced how fundamental this software is? Let me help with this. What the novel contribution of this software for someone or for a corporate to buy or invest in? What are its benefits and how these benefits are not already out there?

Response 2: The novel contributions are clearly stated in the Conclusions section of the revised paper.

“The proposed solution is a fully-functional vertical search engine that works not only on products, but also on other types of extractable information (e.g., food recipes). A novel contribution that this paper presents is a product retrieval solution, which does not depend on XML feeds provided by the retailers, but rather provides a concise, flexible and efficient method of retrieving and extracting information by employing a novel template used to represent the location of the extractable data from web pages. Another novel aspect is the cost-effective method of storing data on a cloud platform and indexing it to minimize resource usage, while also providing efficient solutions for reducing resource consumption and, therefore, the costs in the other system components. Finally, a discussion is presented on the ethics and security that the proposed system poses.”

From an investment point of view, the proposed solution has a great potential for investment. Imagine having a product search engine that does not depend on XML feeds, which, in turn, may not always be up to date (I have seen this happen) and the retailer can, at any time, deny access to those feeds. The product details that are extracted are exactly what the user sees on that retailer’s website.

As far as I have researched, all the product search engines that can be found on the web depend on XML feeds from the retailers.

The proposed solution is to product information as Google search engine is to words, i.e., Google does not need from websites a mapping from each word to each web page it appears in, but rather Google computes that mapping.

Point 3: What is your benchmark test for the optimization? Can you show us some Figures?

Response 3: As stated throughout the paper, the benchmark is the research from [8], and the optimizations are made in order to minimize resource usage. The performance evaluation for some optimizations (e.g., the memcache optimization) is difficult to perform because the Google Cloud Platform manages the web app instances and task queues, and it offers little information and no control over the management process. For the other optimizations, discussions and values are provided in the corresponding sections of the paper. In terms of results and figures, the data is presented in Figures 3 and B1, and Tables A1 and A2.